# From Malicious to Marvelous: The Art of Adversarial Attack as Diffusion

## Abstract

The ubiquitous presence of adversarial attacks in deep learning has been a source of frustration and challenge for researchers for years. However, in this work, we establish a new connection between adversarial attacks and the intricate process of diffusion. Specifically, we formulate an adversarial attack as a diffusion process, and by reverting this adversarial attack process, we have devised an innovative defense mechanism that stands out as a general-purpose defense against both black-box and white-box attacks. We call this new mechanism a Reverse Adversarial Process (RAP), which is ensured by a theoretical treatment for deploying denoising diffusion models on arbitrary distributions. Empirically, we found our model successfully defends against adversarial attacks with an unprecedented level of accuracy. For example, our approach has demonstrated exceptional performance on the *RobustBench*, a highly-regarded leaderboard for assessing adversarial robustness, outperforming previous state-of-the-art methods by a clear margin.

## 1 Introduction

The phenomenon known as adversarial attacks represents a malicious strategy in which attackers manipulate input data in order to confound machine learning algorithms (Goodfellow et al., 2014), rendering them susceptible to erroneous outcomes. Its existence serves as a time bomb, undermining the very foundations of machine learning (Dickson, 16 Dec. 2020). As the utility and prevalence of machine learning continues to expand, including in the development of general artificial intelligence, the specter of adversarial attacks looms larger than ever before. Indeed, the potential consequences of such attacks upon the reliability of machine learning systems are grave and far-reaching.

In recent times, considerable advances have been made in the area of adversarial robustness, as evidenced by the emergence of various techniques and methods, including the heuristic approach known as adversarial training (Kolter, 8 Feb. 2023; Madry et al., 2017; Szegedy et al., 2013; Tramèr et al., 2017). This technique has been demonstrated to be effective in lowering the success rate of adversarial attacks, and thus represents a promising line of defense against these attacks. Despite their empirical successes, it is important to acknowledge that adversarial attacks remain a persistent

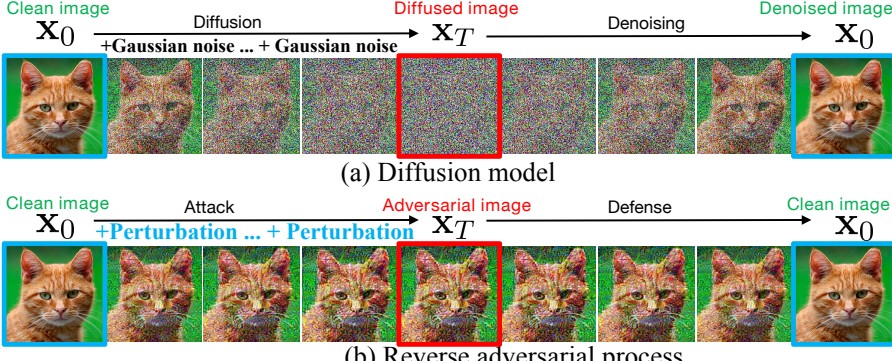

(a) Diffusion model

(b) Reverse adversarial process

Figure 1: A comparison between our reverse adversarial process (RAP) and a diffusion model. Note that, unlike MNIST, adversarial perturbations on CIFAR-10 are generally imperceptible. For visualization purposes, we intentionally amplify the magnitude of the adversarial attack in this figure.

and formidable challenge, and cannot be entirely eliminated through the application of adversarial training alone (Kolter, 8 Feb. 2023; Schott et al., 2018). Additionally, the issue of overfitting (Rice et al., 2020), which is also encountered in the context of adversarial training, represents a significant obstacle that can compromise its effectiveness.

In this new study, we have uncovered a connection between adversarial attacks and the cutting-edge field of diffusion models (see Fig. 1). Our research has revealed that these two seemingly disparate phenomena share a strikingly similar process, with diffusion involving the gradual addition of Gaussian noise to a sample, and adversarial attacks entailing the gradual addition of adversarial perturbations to a sample. This finding has prompted us to explore a new strategy for defending against adversarial attacks, one that draws upon the principles of diffusion modeling. In more technical details, we regard each attack step as a diffusion step, which gradually transforms an initial sample $\mathbf{x}_0$ into an adversarial example $\mathbf{x}_T$ after $T$ steps' perturbation with a sequence of perturbations $\{\xi_1, \cdots, \xi_T\}$ (see Fig. 1(b)). Our goal is to learn a defensive adapter (typically a U-Net (Ronneberger et al., 2015)) to complete its inverse process, i.e., involving a gradual removal of perturbations $\{\xi_T, \cdots, \xi_0\}$ applied to $\mathbf{x}_T$ in a manner that ensures learnability, ultimately leading to the recovery of the original state $\mathbf{x}_0$. Nevertheless, directly applying the diffusion model is impractical due to its requirement of the added perturbation to adhere to a Gaussian distribution. Adversarial perturbations, being a complex distribution [1], presents a challenge. To tackle this, we employ a theoretical deployment that expands the concept of Gaussian noise, typically associated with diffusion models, to encompass adversarial perturbations characterized by intricate distributions. This extension facilitates our incremental denoising approach.

Our approach represents a departure from previous works on adversarial purification using diffusion models (Nie et al., 2022; Ankile et al., 2023; Xiao et al.; Wang et al., 2022; Wu et al., 2022) (see Fig. 2). Recently, researchers explored adding a sequence of Gaussian noises $\{\epsilon_1, \cdots, \epsilon_T\}$ to adversarial examples and used diffusion models to remove these Gaussian noises, hoping "also" to remove the adversarial perturbation. While such approaches are undeniably impressive, they fall short in directly modeling adversarial perturbations, leaving them a room for improvement when defending against white-box attackers who are aware of this denoising process (Nie et al., 2022; Ankile et al., 2023; Xiao et al.; Wang et al., 2022; Wu et al., 2022). Our method, in contrast, directly models adversarial perturbations and can predict adversarial perturbations and remove them. More precisely, our approach can be seen as augmenting a vanilla classifier with an adapter to create a "robust classifier" (see Fig. 3). Notably, our method does not rely on adversaries of the vanilla classifier to train the adapter; instead, we utilize adversaries of the "robust classifier" to train it. In this regard, our method mirrors the principles of adversarial training, specifically training the adapter in a diffusion-denoising manner. Consequently, our method is effective against white-box attacks.

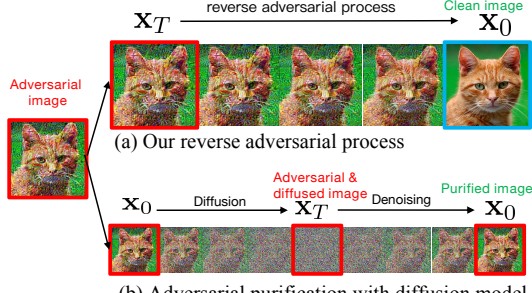

(a) Our reverse adversarial process

(b) Adversarial purification with diffusion model

Figure 2: A comparison between RAP and adversarial purification with a diffusion model. RAP is unique in that it directly models and predicts adversarial perturbations, making it capable of defending against white-box attacks.

This paper presents the following contributions toward enhancing the security and reliability of deep learning systems in the face of adversarial attacks. **Firstly**, we propose a novel approach that directly models and predicts adversarial perturbations, establishing a prominent diffusion learning system that achieves adversarial robustness. **Secondly**, our method is connected to a form of adversarial training that adversarially trains the adapter in a diffusion-denoising way. This unique approach equips our method with the capability to effectively handle perturbations from white-box attacks, thus serving as a robust defense against such attacks. **Thirdly**, our method achieves promising results, surpassing previous state-of-the-art methods on the highly respected *RobustBench* leaderboard for evaluating adversarial robustness in images.

---

[1]If one possesses comprehensive knowledge about the distribution of adversarial perturbations, he can effectively establish a mapping between each adversarial example and its corresponding clean version. However, obtaining an accurate distribution of adversarial perturbations is considerably challenging.

## 2 RELATED WORK

**Adversarial robustness.** Despite the significant amount of research dedicated to tackling the challenge of adversarial robustness, a growing body of evidence suggests that the majority of defenses proposed in the literature fail to offer substantial effectiveness (Athalye et al., 2018; Tramer et al., 2020; Brendel & Bethge, 2017; Carlini et al., 2019; Athalye & Carlini, 2018; Carlini & Wagner, 2017). These failed but pioneering efforts include manifold transformation (Samangouei et al., 2018; Shen et al., 2017; Song et al., 2017; Liao et al., 2018; Meng & Chen, 2017), randomization (Prakash et al., 2018; Dhillon et al., 2018; Xie et al., 2017), shattering gradients (Buckman et al., 2018; Guo et al., 2017; Ma et al., 2018; Kabilan et al., 2021), and using exploding and vanishing gradients (Song et al., 2017; Samangouei et al., 2018).

A identified by Athalye et al. (2018) and Tramer et al. (2020), adversarial training (Rice et al., 2020; Madry et al., 2017; Szegedy et al., 2013; Tramèr et al., 2017) is the only technique widely acknowledged as effective for achieving adversarial robustness. It involves augmenting the training data with adversarial examples. Although adversarial training has shown some successes, it is crucial to recognize that adversarial attacks still pose a persistent and daunting challenge that cannot be completely resolved by relying solely on adversarial training (Kolter, 8 Feb. 2023). Moreover, adversarial training often faces the problem of overfitting (Rice et al., 2020; Schott et al., 2018), which presents a significant obstacle that can hinder its effectiveness. Our method directly models adversarial perturbations, and one of its goals is even to predict adversarial perturbations under white-box attacks. In this sense, our method resembles adversarial training. It is commendable that a work achieved decent adversarial robustness on MNIST (LeCun, 1998) using a generative classifier (Schott et al., 2018). Unfortunately it did not demonstrate effectiveness on natural images such as CIFAR-10 (Krizhevsky et al., 2009).

**Adversarial purification with diffusion model.** We have introduced this group of literature (Nie et al., 2022; Ankile et al., 2023; Xiao et al.; Wang et al., 2022; Wu et al., 2022; Wang et al., 2023) in the introduction section, whereby we have concluded that adversarial purification alone cannot guarantee adversarial robustness. The practice of introducing Gaussian noise and utilizing diffusion models to eliminate it does not guarantee the removal of adversarial perturbations, leaving vulnerabilities to white-box attackers. In contrast, our approach focuses on modeling adversarial perturbations. To achieve this, we enhance a vanilla classifier with an adapter, thereby creating a "robust classifier." Crucially, we train the adapter using adversaries of the "robust classifier," rather than that of the vanilla classifier. This methodology aligns with the principles of adversarial training and equips our method to effectively defend against white-box attacks. The outstanding paper (Yoon et al., 2021) provides supporting evidence for the originality of our work. In its Sec. 3.3, it is stated, "As we will show in Section 5, the defense method based on the deterministic purification described in Section 3.2 can successfully defend most of the adversarial attacks, but it is vulnerable to the strong attack based on the gradient estimation of the full purification process." Compared to prior diffusion-based methods, our approach not only eliminates the forward diffusion process but also formulates adversarial perturbation as a diffusion process in which the adversarial perturbations can be eliminated through a Reverse Adversarial Process.

**Denoising for adversarial robustness.** Some arts attempt to improve model adversarial robustness by denoising, mainly including two technical routes. The first route is to smooth out adversarial perturbations through smoothing filters. However, if the attacker knows about the existence of these filters, filtering techniques alone (Xie et al., 2019; Vuyyuru et al., 2020) are ineffective and may even weaken the model's adversarial robustness, as observed in some works (Wang et al., 2020). The second route is to learn a denoiser to remove adversarial perturbations from adversarial examples. However, previous work has had difficulty learning a generalizable denoiser to remove adversarial perturbations (Li et al., 2021; Jing, 2022; Liao et al., 2018; Creswell & Bharath, 2018), especially in the case of white-box attacks. This is because the distribution of adversarial perturbations is complex and challenging to predict in a single step. Drawing inspiration from diffusion models, we train a denoiser that progressively eliminates adversarial perturbations. We theoretically extend the concept of Gaussian noise, typically employed in diffusion models, to encompass adversarial perturbations with intricate distributions, enabling our step-by-step denoising process.

**Non-Gaussian diffusion model.** Diffusion models are based on the ubiquitous Gaussian distribution, which elegantly relates the states observed at different times by a noise sampled from this distribution.

Extending this model to arbitrary distributions has proven to be a challenge. While recent work has made strides in this direction (Nachmani et al., 2021; Deasy et al., 2021; Deasy, 2022), it falls short of fully characterizing the complexity of real-world distributions. The challenge of adversarial robustness and the pernicious problem of robust overfitting underscore the fact that adversarial perturbations arise from intricate distributions. In the face of this challenge, we not only provide a theoretical deployment, but also validate the effectiveness of our method through extensive experimentation.

## 3 METHOD

In this section, we describe how we view adversarial attacks as a diffusion process (Sec. 3.1). Then, we introduce the reverse adversarial process with denoising score matching (Sec. 3.2) and our training objectives. We also discuss the connection to adversarial training (Sec. 3.3).

### 3.1 ADVERSARIAL ATTACK AS DIFFUSION PROCESS

Taking an image $\mathbf{x}_0$ sampled from the underlying data distribution $p_{data}(\mathbf{x}_0)$ as a starting point, the adversarial attack is a process of gradually applying a sequence of adversarial perturbations $\{\xi_1, \cdots, \xi_t, \cdots, \xi_T\}$ $(0 \le t \le T)$ to $\mathbf{x}_0$. The output of an adversarial attack is the perturbed image $\mathbf{x}_T$ after a given number of steps denoted as $T$ (typically, $T$ is 20, 100, or larger). Noticing that various adversarial attack techniques have been proposed to achieve better attacks, here, for the ease of notation, we define the adversarial perturbation iteration as a step function over image $\mathbf{x}$:

$$\mathbf{x}_{t+1} = \mathrm{s}(\mathbf{x}_t) = \alpha \nabla \mathcal{L}_\phi(\mathbf{x}_t) + \mathbf{x}_t, \quad (1)$$

where $\alpha$ is the step size. We include a noise at the first step which makes $\hat{\mathbf{x}}_0 = \mathbf{x}_0 + \epsilon$ the starting point instead of the original image $\mathbf{x}_0$ and $\epsilon$ is noise sampled from a gaussian distribution $\epsilon \sim \mathcal{N}(0, \sigma\mathbf{I})$ to improve the diversity of the adversary, which is considered

---

**Algorithm 1** Adversarial attack process in RAP

1: Input: a clean example $\mathbf{x}_0$, attack process length $T$
2: **for** t= 0, ..., T **do**
3:     **if** t=0 **then**
4:         $\zeta \sim \mathrm{Uniform}(-c, c)$
5:         $\mathbf{x}_0 \leftarrow \mathbf{x}_0 + \zeta$
6:     **end if**
7:     $\xi_t \leftarrow \mathrm{Sign}(\frac{\partial \mathrm{L}}{\partial \mathbf{x}}\big|_{\mathbf{x}=\mathbf{x}_{t-1}})$,
8:     $\mathbf{x}_t \leftarrow \mathrm{Clamp}(\mathbf{x}_{t-1} + \lambda\xi_t)$
9: **end for**
10: **return** an adversary $\mathbf{x}_T$

---

standard in adversarial attack (Madry et al., 2017; de Jorge Aranda et al., 2022). Therefore, we have the static mapping between the random variables $\mathbf{x}_t = \mathrm{s}^t(\hat{\mathbf{x}}_0)$. $\mathcal{L}$ is an adversarial loss function (e.g., an adversarial classification loss) taking the classifier $\phi$ and current perturbation $\mathbf{x}_t$ as an input.

Here we introduce the assumption that the step function is invertible. Although in reality, this assumption may not hold, we empirically find our algorithm works well and will not explicitly use this assumption. Thus, our algorithm is more general than the analysis here.

For the adversarial process we care about, we can get their marginal distribution. Thanks to the invertibility of the step function, we have $\hat{\mathbf{x}}_0 = \mathrm{s}^{-t}(\mathbf{x}_t)$:

$$p(\mathbf{x}_t \mid \mathbf{x}_0) = \mathcal{N}(\mathrm{s}^{-t}(\mathbf{x}_t) \mid \mathbf{x}_0, \sigma\mathbf{I}) \cdot \det\left(\frac{d\mathrm{s}^{-t}}{d\mathbf{x}}(\mathbf{x}_t)\right). \quad (2)$$

Then the adversarial distribution at each time step is denoted as:

$$p(\mathbf{x}_t) = \int p_{data}(\mathbf{x}_0)p(\mathbf{x}_t|\mathbf{x}_0)d\mathbf{x}_0. \quad (3)$$

In summary, an adversarial attack can be regarded as a diffusion process of degrading an original natural image into an adversary. Fig. 1(b) shows an overview of this process, and the detailed algorithm is presented in Alg. 1.

### 3.2 REVERSE ADVERSARIAL PROCESS AS DEFENCE

The adversarial process involves a finite number of steps to transform data into the adversarial distribution, allowing for the possibility of reversing the process to recover the original data from its adversarial example. This reversibility concept has been demonstrated in various approaches, including the score-based generative model (Song & Ermon, 2019) and denoising diffusion models (Ankile et al., 2023), all of which are discretizations of underlying stochastic differential equations.

In this context, we adopt the score-based perspective and formulate the denoising score matching approach for our reverse adversarial process. Utilizing a reversible step function, we demonstrate that learning the gradient score function is tantamount to acquiring knowledge about the mapping between the adversarial and the original image. This can also be viewed as applying a generalized diffusion approach (Bansal et al., 2022) to address the problem.

In particular, we learn a parameterized function $f_\theta(\mathbf{x}, t)$ to approximate the score of the adversarial distribution at time step $t$:

$$\boldsymbol{\theta}^* = \arg\min_{\boldsymbol{\theta}} \sum_{t=1}^{T} \mathbb{E}_{p_{\mathrm{data}}(\mathbf{x}_0)} \mathbb{E}_{p(\mathbf{x}_t|\mathbf{x}_0)} \left[ \| f_\theta(\mathbf{x}_t, t) - \nabla \log p(\mathbf{x}_t \mid \mathbf{x}_0) \|_2^2 \right], \qquad (4)$$

where notably, the last term in Equation 4 is:

$$\nabla \log p(\mathbf{x}_t \mid \mathbf{x}_0) = \nabla \log \mathcal{N}(\mathrm{s}^{-t}(\mathbf{x}_t) \mid \mathbf{x}_0, \sigma\mathbf{I}) + \nabla \log \det \left( \frac{d\mathrm{s}^{-t}}{d\mathbf{x}}(\mathbf{x}_t) \right)$$

$$= -\frac{\mathrm{s}^{-t}(\mathbf{x_t}) - \mathbf{x}_0}{\sigma^2} \cdot \frac{d\mathrm{s}^{-t}}{d\mathbf{x}}(\mathbf{x}_t) + \nabla \log \det \left( \frac{d\mathrm{s}^{-t}}{d\mathbf{x}}(\mathbf{x}_t) \right)$$

Due to our assumption that the step function is invertible, the Jacobian $\frac{d\mathrm{s}^{-t}}{d\mathbf{x}}(\mathbf{x}_t)$ is also invertible. We use the notation $\mathbb{J}_{\mathbf{x}_t}$ to further compress the notation. Minimizing the denoising score matching objective is equivalent to minimizing:

$$\| f_\theta(\mathbf{x}_t, t) - \nabla \log p(\mathbf{x}_t \mid \mathbf{x}_0) \|_2^2 = \left\| \left[ (f_\theta(\mathbf{x}_t, t) - \nabla \log \det \mathbb{J}_{\mathbf{x}_t}) \mathbb{J}_{\mathbf{x}_t}^{-1} + \mathrm{s}^{-t}(\mathbf{x}_t) - \mathbf{x}_0 \right] \frac{\mathbb{J}_{\mathbf{x}_t}}{\sigma^2} \right\|_2^2, \qquad (5)$$

$$\leq \frac{\lambda_{max}(\mathbb{J})}{\sigma^2} \left\| (f_\theta(\mathbf{x}_t, t) - \nabla \log \det \mathbb{J}_{\mathbf{x}_t}) \mathbb{J}_{\mathbf{x}_t}^{-1} + \mathrm{s}^{-t}(\mathbf{x}_t) - \mathbf{x}_0 \right\|_2^2, \qquad (6)$$

where $\lambda_{\max}(\mathbb{J})$ represents the maximum eigenvalue of the Jacobian matrix calculated for all $\mathbf{x}_t$. The upper bound analysis demonstrates that we can minimize an upper bound directly. Additionally, we've observed that the majority of terms within the $l_2$ norm primarily rely on $\mathbf{x}_t$. Rather than modeling the gradient of the score function, we introduce a new parameterization to learn all the terms dependent on $\mathbf{x}_t$ within the $l_2$ norm, resulting in a simplified loss term:

$$\tilde{f}_\theta(\mathbf{x}_t, t) = (f_\theta(\mathbf{x}_t, t) - \nabla \log \det \mathbb{J}_{\mathbf{x}_t}) \mathbb{J}_{\mathbf{x}_t}^{-1} + \mathrm{s}^{-t}(\mathbf{x}_t). \qquad (7)$$

Getting back to our objective function, we now have:

$$\boldsymbol{\theta}^* = \arg\min_{\boldsymbol{\theta}} \sum_{t=1}^{T} \mathbb{E}_{p_{\mathrm{data}}(\mathbf{x}_0)} \mathbb{E}_{p(\mathbf{x}_t|\mathbf{x}_0)} \left\| \tilde{f}_\theta(\mathbf{x}_t, t) - \mathbf{x}_0 \right\|^2. \qquad (8)$$

**Algorithm 2** Reverse adversarial process in RAP

1: Input: an adversary $\mathbf{x}_T$, defence process length $T$
2: **for** t= T, ..., 1 **do**
3: $\quad \hat{\mathbf{x}}_0 \leftarrow \tilde{f}_\theta(\mathbf{x}_t, t)$
4: $\quad$ using Alg. 1 to compute $\xi_{t-1}$ and $\mathbf{x}_{t-1}$ by inputting $(\hat{\mathbf{x}}_0, t-1)$
5: **end for**
6: **return** a clean image $\mathbf{x}_0$

We notice in the literature that this approach is closely related to the generalized diffusion method proposed by Bansal et al. (Bansal et al., 2022). Empirical evidence from their work suggests that this formalism possesses the generality to reverse any diffusion process.

After learning our generalized diffusion model $f_\theta$, we construct the reverse adversarial process. Given the intermediate adversary $\mathbf{x}_t$, we sample $\mathbf{x}_{t-1}$ using the following expression:

$$p(\mathbf{x}_{t-1} \mid \mathbf{x}_t) = p(\mathbf{x}_{t-1} \mid \tilde{f}_\theta(\mathbf{x}_t, t)) = \mathrm{s}^{t-1}(\tilde{f}_\theta(\mathbf{x}_t, t) + \epsilon), \qquad (9)$$

which is basically navigating the adversarial process to predict $\mathbf{x}_0$.

In summary, a reverse adversarial process can be regarded as a gradual process of removing adversarial perturbations and restoring an original clean image. Fig. 2(a) presents an overview of this process, and Alg. 2 details the algorithm.

### 3.3 CONNECTION TO ADVERSARIAL TRAINING

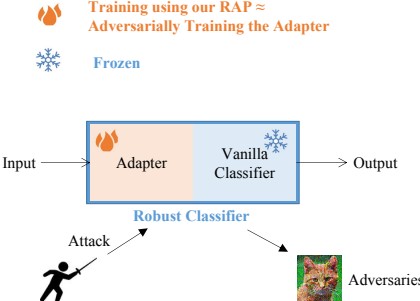

Figure 3: Connection to adversarial training.

What's particularly intriguing about our approach is that it involves enhancing a conventional classifier by incorporating an adapter (i.e., $\tilde{f}_\theta$ in Alg. 2 and Eqn. 8), resulting in the creation of a "robust classifier" (see Fig. 3). Notably, our method doesn't depend on adversaries of the vanilla classifier for adapter training; instead, we utilize adversaries of the full "robust classifier." To elaborate, we employ Alg. 1 to attack the "robust classifier", generating adversarial examples $\mathbf{x}_t$. We then feed these adversarial examples into the adapter to obtain $\hat{\mathbf{x}}_0$. In this context, our approach aligns with the principles of adversarial training, particularly by training the adapter using a diffusion-denoising approach. As a result, our method can be effective in defending against white-box attacks.

### 3.4 TRAINING

The training process of RAP is similar to that of a diffusion model, but there are two notable differences. First, the diffusion process in RAP is obtained through the adversarial attack process in Alg. 1. Second, in the first step of the diffusion process in RAP, random perturbations are introduced. This is inspired by adversarial training (Madry et al., 2017), and it has been shown to be beneficial in practice. Third, we have a classification loss for $\hat{\mathbf{x}}_0$. To train a diffusion model, there are two forms of targets to choose from, namely noise and original images. Empirically, the difference in effectiveness between these two targets is not significant. In this paper, we choose the original images as the target (see Alg.3).

---

**Algorithm 3** RAP training process.

1: **Input:** dataset $d$, diffusion steps $T$, noise schedule $\sigma_1, ..., \sigma_T$, label $y$.
2: **repeat**
3:      $\mathbf{x}_0 \sim d(\mathbf{x}_0)$
4:      $t \sim \mathrm{U}(\{1, ..., T\})$
5:      using Alg. 1 to compute $\xi_t$ and $\mathbf{x}_t$ by inputting $(\mathbf{x}_0, t)$
6:      Take gradient descent step on the loss: $\|\mathbf{x}_0 - \tilde{f}_\theta(\mathbf{x}_t, t)\| + CE(\tilde{f}_\theta(\mathbf{x}_t, t), y)$
7: **until** converged

---

## 4 EXPERIMENTS

In this section, we start with the experimental setup (Sec. 4.1), followed by presenting quantitative results on the reverse adversarial process's efficacy in removing adversarial perturbations (Sec. 4.2). Next, we compare our method to state-of-the-art approaches (Sec. 4.3). Finally, we perform an ablation study to understand the roles of its components (Sec. 4.4).

### 4.1 EXPERIMENTAL SETUP

**Datasets.** We mostly focus on CIFAR-10 (Krizhevsky et al., 2009), which are currently the most widely adopted datasets for evaluating adversarial robustness, particularly in the context of white-box attacks. We utilize the MNIST dataset (LeCun, 1998) to examine the generality of our method.

**Classifiers.** We conduct experiments by using the same architectures as prior arts. To be more specific, for CIFAR-10, our method runs using the identical architecture as our competitors, i.e., wide-resnet-28-10 (Zagoruyko & Komodakis, 2016). For the MNIST dataset, we adopt a minimalistic CNN model [2], which is readily available as an example in the PyTorch library (Paszke et al., 2019).

**Threat models.** Here, we highlight the following key information.

- **Attack methods.** We evaluated the robustness of our method against strong adaptive attacks. As a standard practice, RobustBench (Croce et al., 2020) [3], a highly regarded leaderboard for adversarial robustness assessment, is employed for the majority of experiments (excluding Table 3 and Table 5). More specifically, we utilize the strong AutoAttack (Croce & Hein, 2020) to perform experiments and present the outcomes, which includes both white-box and *black-box*. For Table 5, we examine BPDA(Athalye et al., 2018) and EoT(Athalye et al., 2018; Hill et al., 2020). In addition to these

---

[2] https://github.com/pytorch/examples/tree/main/mnist
[3] https://robustbench.github.io/

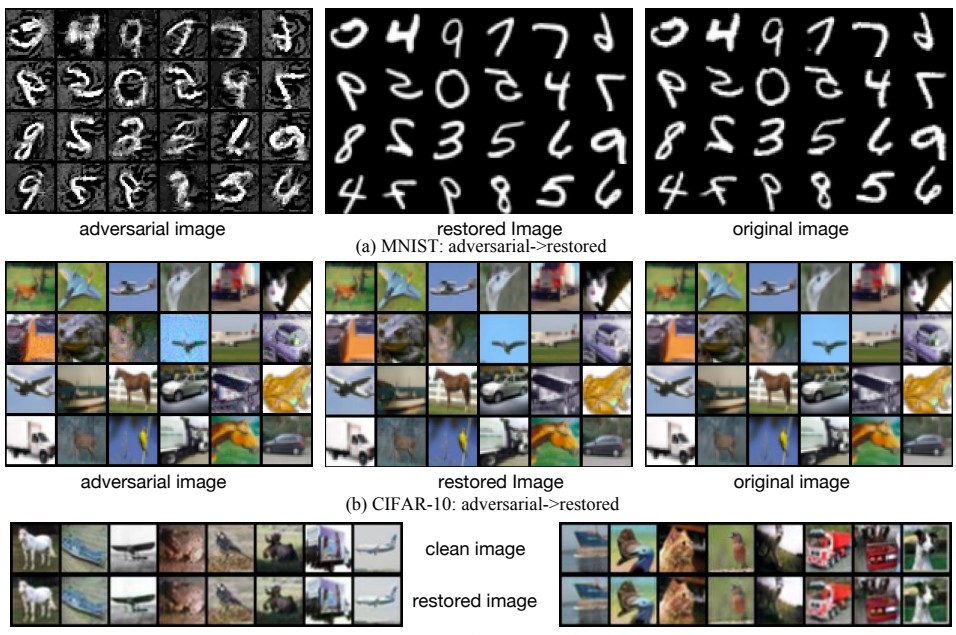

Figure 4: Visualization of adversarial images, restored images, and original clean images. More visualization results are in the Appendix.

strong adaptive attacks, we also test our method against several classical attacks, such as DeepFool (Moosavi-Dezfooli et al., 2016), BIM (Kurakin et al., 2018), and FGSM (Goodfellow et al., 2014).

- **No randomness.** It is well known that the introduction of randomness can lead to an overestimation of the network's robustness (Athalye et al., 2018). To eliminate the influence of randomness on the evaluation, in the defense stage, we **deactivate** Langevin dynamics, ensuring that the entire inference process is free from any randomness.

- **Imperceptibility.** We assessed the performance of our method against both $\ell_\infty$ attacks and $\ell_2$ attacks. For $\ell_\infty$ attacks, the maximum attack magnitudes are 0.3 on MNIST and 8/255 on CIFAR-10. For $\ell_2$ attacks, the maximum attack magnitude are 1.5 on MNIST and 0.5 on CIFAR-10.

**Evaluation metrics.** We utilize two metrics to evaluate the effectiveness of each defense method: standard accuracy and robust accuracy. Standard accuracy refers to the prediction accuracy of each defense method before an attack, while robust accuracy refers to the prediction accuracy after an attack. We have found that attacking our RAP model is exceedingly difficult for adaptive attack methods, costing a significant computational cost. Therefore, we cannot afford to perform attacks on the entire dataset to evaluate our method. (Nie et al., 2022) faced the same challenge and opted for a fixed set of 512 images for evaluation. In line with their approach, we use the exact set of images obtained from their official GitHub repository. It's worth noting that the performance of the compared methods on this fixed set generally aligns with their performance on the complete dataset.

### 4.2 VISUALIZATION RESULTS

Given that our method pioneers the use of a diffusion model to model adversarial perturbations, and considering that the distribution of these perturbations deviates from the assumption of Gaussian noise in the standard diffusion model, one may be understandably eager to know if our trained RAP model can truly eliminate adversarial perturbations and faithfully reconstruct the original image, or whether the model can predict the adversarial noise as closely as possible.

The qualitative results presented in Fig. 4, 5, and 6 demonstrate the effectiveness of our method. It can reliably restore high-quality original images even when faced with substantial adversarial perturbations. This highlights our method's capability to learn robust and adaptable representations, overcoming challenges posed by non-Gaussian adversarial perturbation distributions and achieving good quality in generation and restoration tasks.

Table 1: Standard accuracy and robust accuracy against AutoAttack $\ell_\infty$ ($\epsilon$ = 8/255) on CIFAR-10 on the *RobustBench*, obtained by different classifier architectures. Results are sorted by robust accuracy.

| Method | Extra Data | Standard Acc | Robust Acc |
|---|---|---|---|
| Ours | ✗ | **94.53** | **73.24** |
| (Nie et al., 2022) | ✗ | 90.07 | 71.29 |
| (Wang et al., 2023) | ✓ | 93.25 | 70.69 |
| (Rebuffi et al., 2021) | ✓ | 92.23 | 68.56 |
| (Wang et al., 2023) | ✓ | 92.44 | 67.31 |
| (Gowal et al., 2021) | ✗ | 88.74 | 66.60 |
| (Gowal et al., 2020) | ✓ | 91.10 | 66.02 |
| (Huang et al., 2022) | ✗ | 91.58 | 65.79 |
| (Rebuffi et al., 2021) | ✗ | 88.54 | 64.46 |
| (Kang et al., 2021) | ✓ | 93.73 | 71.28 (64.20) |
| (Xu et al., 2023) | ✗ | 93.69 | 63.89 |
| (Pang et al., 2022) | ✗ | 89.01 | 63.35 |
| (Sehwag et al., 2021) | ✗ | 87.30 | 62.79 |
| (Wu et al., 2020) | ✓ | 88.25 | 62.11 |
| (Zhang et al., 2020) | ✓ | 89.36 | 59.96 |
| (Gowal et al., 2020) | ✗ | 85.29 | 59.57 |
| (Wu et al., 2020) | ✗ | 85.36 | 59.18 |

Table 2: Standard accuracy and robust accuracy against AutoAttack $\ell_2$ ($\epsilon$ = 0.5) on CIFAR-10 on the *RobustBench*, obtained by different classifier architectures. Results are sorted by robust accuracy.

| Method | Extra Data | Standard Acc | Robust Acc |
|---|---|---|---|
| Ours | ✗ | **95.76** | **85.54** |
| (Wang et al., 2023) | ✓ | 95.54 | 84.97 |
| (Wang et al., 2023) | ✓ | 95.16 | 83.68 |
| (Rebuffi et al., 2021) | ✓ | 95.74 | 82.32 |
| (Rebuffi et al., 2021) | ✗ | 92.41 | 80.86 |
| (Nie et al., 2022) | ✗ | 92.68 | 80.60 |
| (Gowal et al., 2020) | ✓ | 94.74 | 80.53 |
| (Augustin et al., 2020) | ✓ | 93.96 | 78.79 |
| (Sehwag et al., 2021) | ✗ | 90.93 | 77.24 |
| (Wu et al., 2020) | ✓ | 92.23 | 76.25 |
| (Gowal et al., 2020) | ✗ | 90.90 | 74.50 |
| (Wu et al., 2020) | ✗ | 88.51 | 73.66 |
| (Augustin et al., 2020) | ✗ | 91.08 | 72.91 |
| (Ding et al., 2018) | ✗ | 88.02 | 67.77 |
| (Rony et al., 2019) | ✗ | 89.05 | 66.41 |

Particularly, when a clean image is provided as input, the resulting images exhibit qualitative similarities (see Fig. 4(c) and Fig. 7). Line 3 of Alg. 2 effectively accounts for this behavior. When $x_t$ is input into the network, it accurately predicts a clean $x_0$. Similarly, when a clean $x_0$ is given as input, the network consistently outputs another clean $x_0$.

### 4.3 BENCHMARKING THE STATE OF THE ART

**CIFAR-10.** First of all, we compare our RAP method with the current state-of-the-art approaches on *RobustBench*, a widely recognized leaderboard for evaluating adversarial robustness.

Table 1 displays our method's performance on *RobustBench* against AutoAttack, using a perturbation magnitude of $\ell_\infty$ = 8/255. From Table 1, we can make two important observations. Firstly, our method performs well on *RobustBench*, positioning itself as the first place. This emphasizes the value of our approach in modeling and predicting adversarial perturbations. Secondly, our method achieves remarkable standard accuracy, surpassing the second-ranked competitor by a substantial margin. This shows the potential for achieving high robust accuracy and standard accuracy simultaneously.

Table 2 presents the results depicting our method's performance against AutoAttack on *RobustBench*, with an $\ell_2$ perturbation magnitude of 0.5. Just as in Table 1, we observe similar results in Table 2, reaffirming the effectiveness of our method.

Regarding adversarial purification methods, we have made comparisons with DiffPure (Nie et al., 2022) and (Wang et al., 2023) in Tables 1 and 2. Additionally, in Table 5, we further compare our method to (Nie et al., 2022) and (Hill et al., 2020). These comparisons highlight the effectiveness of our method.

RobustBench stands as one of the most reliable leaderboards for evaluating adversarial robustness, employing an ensemble of diverse attacks, with its employed auto-attack being recognized as one of the most potent adversaries. We are guided by prior research (Nie et al., 2022) and primarily concentrate on assessing performance within the RobustBench environment. Additionally, it's worth noting that we have extended our analysis beyond the scope of RobustBench. Supplementary results pertaining to CIFAR-10 have been provided in our Appendix. In Tables 5, the presented data exemplifies the superior performance of our method in comparison to existing approaches.

Table 3: Results for different models, adversarial attacks and distance metrics on ImageNet. Each entry the model's accuracy against adversarial perturbations bounded by the thresholds $\epsilon_{\ell_2}$ = 1.5 and $\epsilon_{\ell_\infty}$ = 0.3. *DeepFool (Moosavi-Dezfooli et al., 2016), BIM (Kurakin et al., 2018), FGSM (Goodfellow et al., 2014).*

| | | CNN | Binary CNN | Madry et al. | Ours |
|---|---|---|---|---|---|
| | Clean | 99.1 | 98.5 | 98.8 | **99.0** |
| $\ell_2$ | Gaussian Noise | 96 | 92 | 96 | **98.9** |
| | DeepFool | 18 | 11 | 91 | **97.3** |
| | $\ell_\infty$ BIM | 13 | 11 | 88 | **93.5** |
| $\ell_\infty$ | FGSM | 4 | 77 | 93 | **94.8** |
| | $\ell_\infty$ DeepFool | 0 | 74 | 90 | **92.3** |
| | BIM | 0 | 70 | 90 | **94.1** |

**MNIST.** We utilize the MNIST dataset to examine the generality of our method. The adversarial attacks we employed encompass DeepFool, BIM, and FGSM. The respective iteration numbers for DeepFool and BIM are 50 and 20, while the corresponding step sizes are 0.02 and 2/255. Our

method's robust performance against various perturbations on the MNIST dataset is presented in Table 3. A closer look at the table reveals that our method outperforms the competitors. These results support the effectiveness of our method against threat models tailored for the MNIST dataset, underscoring the value of our method in modeling adversarial perturbations.

### 4.4 Ablation Analysis

**Generalization and transferability.** One notable limitation of adversarial training is its inherent challenge in achieving generalization(Laidlaw et al., 2020). Specifically, a model trained to withstand one type of attack may struggle to effectively defend against another type of attack. Hence, when considering generalization and transferability, it becomes imperative to compare our method against adversarial training. The findings presented in Table 4 compellingly demonstrate that our method exhibits a substantially superior capacity for generalization compared to adversarial training.

Please refer to the Appendix for comprehensive ablation studies providing deeper insights.

Table 4: Evaluation of generalization and transferability on CIFAR-10 (RobustBench, i.e., employing the strong AutoAttack technique to yield results). Evaluation of generalization and transferability on CIFAR-10. *AT ($\ell_\infty$) (Laidlaw et al., 2020), AT ($\ell_2$) (Laidlaw et al., 2020), PAT-self (Laidlaw et al., 2020),* Adv. Craig *(Dolatabadi et al., 2022),* Adv. GradMatch *(Dolatabadi et al., 2022), DiffPure (Nie et al., 2022).*

| Method | Standard Acc | Robust Acc | |
|---|---|---|---|
| | | $\ell_\infty$ | $\ell_2$ |
| AT ($\ell_\infty$) | 86.8 | 49.0 | 19.2 |
| AT ($\ell_2$) | 85.0 | 39.5 | 47.8 |
| PAT-self | 82.4 | 30.2 | 34.9 |
| Adv. Craig | 83.2 | 40.0 | 33.9 |
| Adv. GradMatch | 83.1 | 39.2 | 34.1 |
| DiffPure | 88.2 | 70.0 | 70.9 |
| Ours ($\ell_\infty$) | **94.5** | **73.2** | **80.2** |
| Ours ($\ell_2$) | **95.8** | **70.2** | **85.5** |

### 4.5 Analysis of Computational Cost

The computational cost of training the RAP model is reasonably comparable to other adversarial attack/defense methods. First, in contrast to adversarial training methods, our approach focuses solely on training the RAP module for 50 epochs, whereas typical adversarial training methods involve training classifiers with adversarial examples for 200 epochs. Second, comparatively, the diffusion time step used in our methods is 20, which facilitates easier learning compared to prior purification methods utilizing diffusion models with a time step of 1,000.

Similarly, the inference time of the RAP model is not significantly more computationally expensive than other adversarial attack/defense methods. First, though our method includes an extra RAP model when compared to adversarial training methods, we can perform a DDIM-like process of RAP inference. As a result, the total inference time with and without RAP is 7.35 images/second and 7.54 images/second (1-step RAP is measured, using GPU A6000 while the machine runs other programs), respectively. This comparison verifies that the inference time of our method is not significantly more demanding than adversarial training methods. Second, moreover, in comparison to purification methods employing diffusion models, our inference time step can be reduced to merely 1, while purification methods with diffusion models entail an inference time step of at least 50. This suggests that our method is dozens of times faster than purification methods utilizing diffusion models.

## 5 Conclusion and Highlight

In this paper, we present a novel approach to address adversarial perturbation modeling. By considering adversarial attacks as a diffusion phenomenon, we introduce a reverse adversarial process that effectively eliminates adversarial perturbations while exhibiting exceptional generalization capabilities. Leveraging our RAP methodology enables the training of expansive models utilizing vast datasets. These models have the potential to serve as foundational pillars for enhancing adversarial robustness across diverse domains of society. As a result, we are committed to conducting extensive research on these formidable models in the future.

**Highlight.** By training an adapter for an existing classifier, our method can enhance adversarial robustness. Therefore, with RAP, it could be feasible to develop a "Foundation Adapter" for numerous existing classifiers, serving as a foundational model for adversarial robustness and providing security across various domains in society.

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

## A   MORE VISUALISATION RESULTS

Qualitative results are presented in Fig. 4, illustrating the effectiveness of our method. We also provide additional qualitative results in Fig. 5 and Fig. 6 to further support our findings. These visuals demonstrate our method's ability to restore original images, even when exposed to large adversarial perturbations. This showcases our method's capacity to learn robust representations, even when dealing with non-Gaussian perturbation distributions. Our method performs well in both generation and restoration tasks, delivering high-quality results despite deviations from Gaussian distributions commonly associated with adversarial perturbations.

Particularly, when a clean image is provided as input, the resulting images exhibit qualitative similarities (see Fig. 7). Line 3 of Alg. 2 effectively accounts for this behavior. When $x_t$ is input into the network, it accurately predicts a clean $x_0$. Similarly, when a clean $x_0$ is given as input, the network consistently outputs another clean $x_0$.

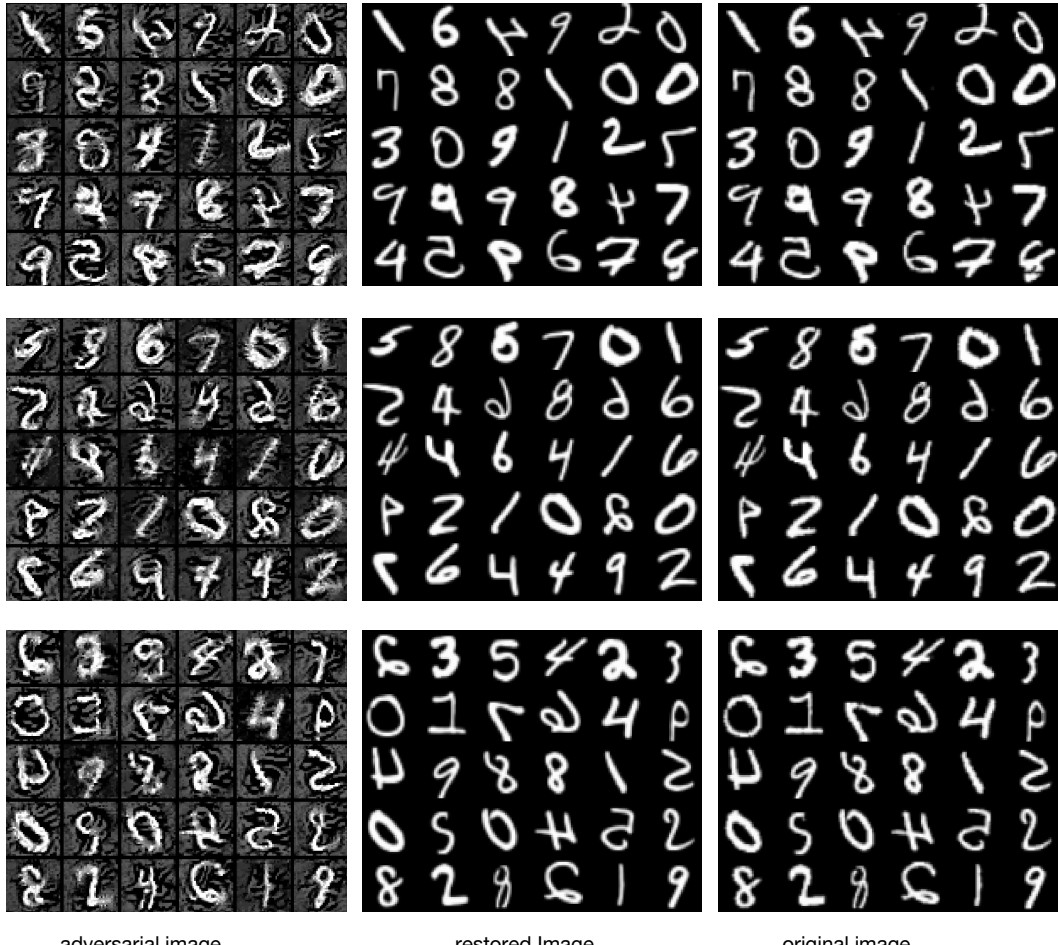

Figure 5: Comparison of adversarial images, restored images, and original clean images on the MNIST dataset.

## B   COMPARISON WITH OTHER PURIFICATION METHODS

Our approach differs from adversarial purification methods as it directly models adversarial perturbations. In the following sections, we empirically compare our method with purification methods. Unlike our technique, many purification-based methods involve computationally expensive optimization processes or sampling loops during the inference stage, making it challenging to subject

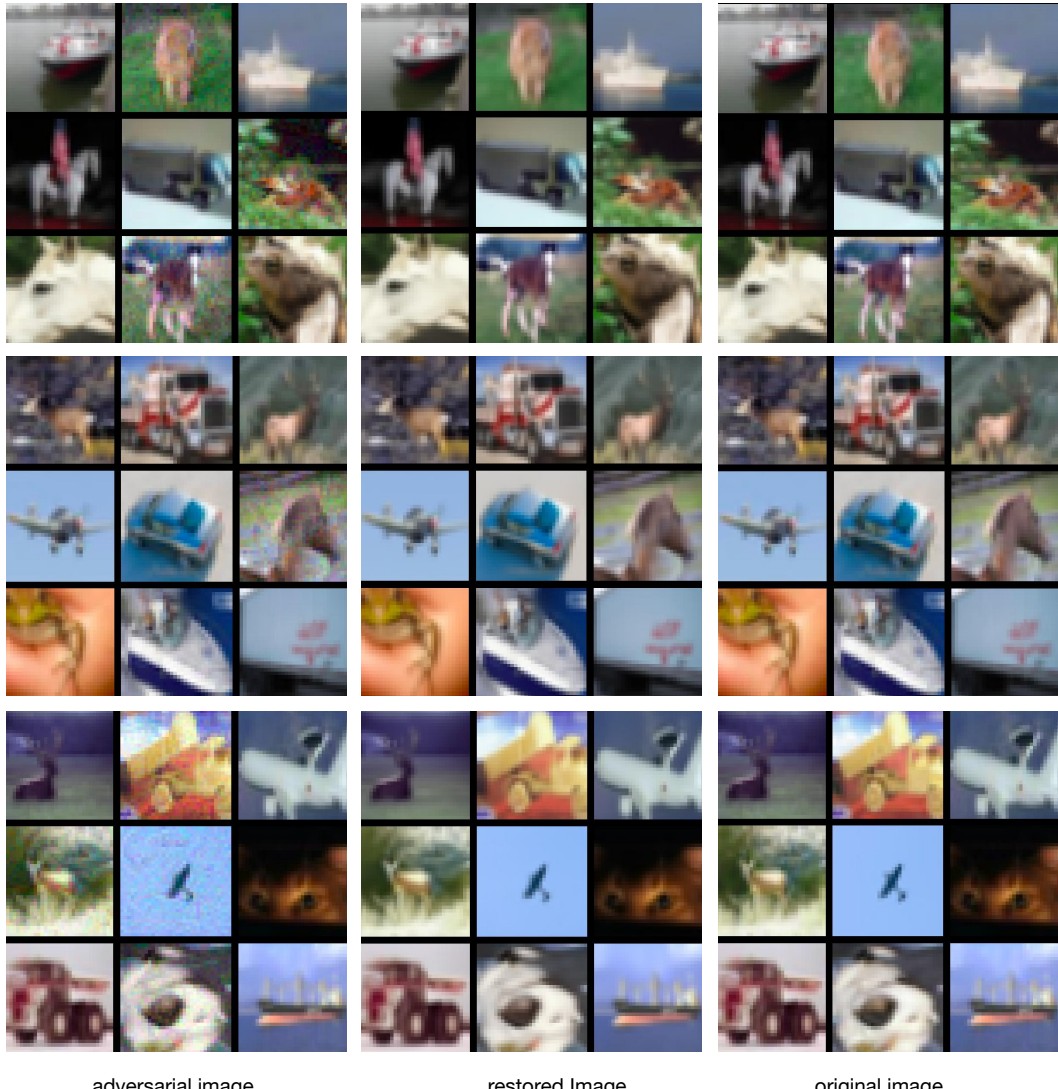

adversarial image          restored Image          original image

Figure 6: Comparison of adversarial images, restored images, and original clean images on the CIFAR-10 dataset.

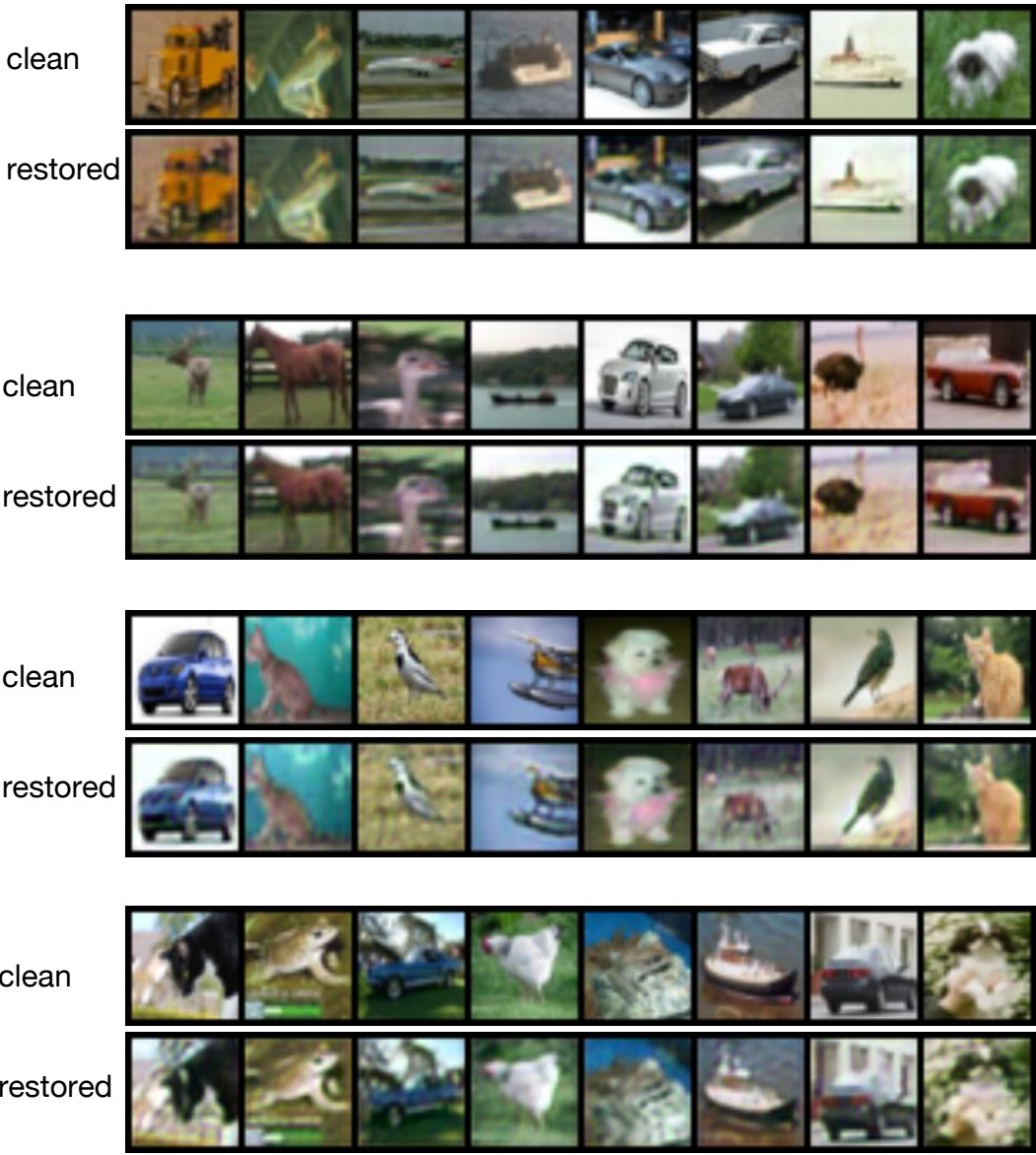

Figure 7: Comparison of the clean images and their associated restored images on the CIFAR-10 dataset.

Table 5: Comparison with other adversarial purification methods on CIFAR-10 using the BPDA+EOT attack with $\ell_\infty$ perturbations. We keep the experimental settings the same with (Hill et al., 2020; Nie et al., 2022;?), where $\epsilon = 8/255$. ($^*$The purification is actually a variant of the Langevin dynamics sampling.) AutoAttack employs a diverse ensemble of attacks, including strong adaptive white-box and black-box attacks, which makes it more powerful than BPDA+EoT. As a result, the attack results in this table are less effective than those presented in Table 1 of the main paper. It's worth noting that the observation of AutoAttack being stronger than BPDA+EoT is consistent with the findings in (Nie et al., 2022) (refer to Table 1 vs. Table 5(f) in (Nie et al., 2022)).

| Method | Purification | Standard Acc | Robust Acc |
|---|---|---|---|
| (Song et al., 2017) | Gibbs Update | 95.00 | 9.00 |
| (Yang et al., 2019) | Mask+Recon. | 94.00 | 15.00 |
| (Hill et al., 2020) | EBM+Langevin dynamics | 84.12 | 54.90 |
| (Yoon et al., 2021) | DSM+Langevin dynamics$^*$ | 86.14 | 70.01 |
| DiffPure (Nie et al., 2022) ($t^* = 0.075$) | Diffusion | 91.03 | 77.43 |
| DiffPure (Nie et al., 2022) ($t^* = 0.1$) | Diffusion | 89.02 | 81.40 |
| Stochastic Security (Hill et al., 2020) | EBM | 84.12 | 78.91 |
| Ours | Diffusion | **94.62** | **81.63** |

Table 6: Combination with adversarial training on CIFAR-10 ($\ell_\infty$ attack). (RobustBench, i.e., employing the strong AutoAttack technique to yield results)

| Method | AT | RAP | RAP+AT |
|---|---|---|---|
| Robust Acc | 49.0 | 73.2 | 73.5 |

them to stronger white-box adaptive attacks, such as AutoAttack. To address this challenge, we employ BPDA+EOT attacks as an evaluation metric, which, although not as potent as AutoAttack, are commonly used in assessing purification methods.

The performance of our method compared to state-of-the-art approaches can be observed in Table 5, providing further evidence of the effectiveness of our technique.

## C    COMBINATION WITH ADVERSARIAL TRAINING

Our method and adversarial training are not in conflict but rather complement each other. This allows us to use the samples purified by our model as input for the adversarially trained model during inference. The results presented in Table 6 demonstrate the additional improvement in robust accuracy achieved through this approach. Notably, when exposed to the $\ell_\infty$ attack, our method's robust accuracy increases from 73.2 to 73.5. This result showcases the effectiveness of our approach.

# D    CODES

```python
class AdversarialDiffusion(nn.Module):
    def __init__(self, ...):
        super().__init__()
        self.linf_attacks = nn.ModuleList(self.get_attacks())
    def get_attacks(self):
        attacks = []
        for i in range(self.num_timesteps):
            attacks.append(JointAttack())
        return attacks
    def q_sample(self, x_start, t, y, cls_model, denoise_fn):
        max_iters = torch.max(t)
        all_attacks = []
        x = x_start
        for i in range(max_iters+1):
            x_raw = x
            if i == 0:
                x_raw = self.linf_attacks[i](x_raw, y, cls_model, denoise_fn, x_start,
                                             rand_start=True)
            else:
                x_raw = self.linf_attacks[i](x_raw, y, cls_model, denoise_fn, x_start)

            x = x_raw.detach()
            all_attacks.append(x)
        all_attacks = torch.stack(all_attacks)
        choose_attack = []
        for step in range(t.shape[0]):
            if step != -1:
                choose_attack.append(all_attacks[t[step], step])
            else:
                choose_attack.append(x_start[step])
        return torch.stack(choose_attack)
    def p_losses(self, x_start, t, y, cls_model, denoise_fn):
        x_attack = self.q_sample(x_start=x_start, t=t, y=y, cls_model=cls_model,
                                 denoise_fn=denoise_fn)
        x_recon = self.denoise_fn(x_attack, t)
        return (x_start-x_recon).abs().mean()+cls_model(
            torch.clamp(x_recon*0.5+0.5, 0, 1),y)
    def forward(self, x, y, model, sampling, *args, **kwargs):
        t = torch.randint(...).long()
        return self.p_losses(x, t, y, cls_model, denoise_fn, *args, **kwargs)
```

