# OpenReview forum: "From Malicious to Marvelous: The Art of Adversarial Attack as Diffusion"
_ICLR.cc/2024/Conference — ICLR 2024 Conference Withdrawn Submission_

### Official Review · Reviewer_dpgy · 2023-10-19

**Soundness:** 1 poor
**Presentation:** 3 good
**Contribution:** 1 poor
**Rating:** 1
**Confidence:** 4

**Summary:**

The paper introduces a novel adversarial purification technique that directly models the process of adversarial modification as a diffusion process. At training time, the method generates adversarial examples using a robust classifier and trains a denoising model that purifies the adversarial examples. At inference time, the method purifies adversarial examples using the trained denoising model and pass the resulting images to a vanilla classifier. The paper claims that the proposed method achieves state-of-the-art performance in terms of both standard and robust accuracy against the strongest AutoAttack.

**Strengths:**

- The presentation is good, particularly the clear overview of the methodology and the effective visualization of images restored through the purification process.
- They consider various types of attack methods, including AutoAttack and BPDA.

**Weaknesses:**

- Recent studies indicate a notable shortcoming of adversarial purification methods, pointing out that most are susceptible to stronger attacks [1, 2]. It seems the authors employ the adjoint method for computing gradients of the diffusion model similar to DiffPure [3]. However, it has been shown that DiffPure is vulnerable to a PGD attack with surrogate gradients [2]. I think the proposed method is also vulnerable to this adaptive attack. It is important to note that most of previous methods on adversarial purification are proven to fall short within six months. Until the authors can show that their method is robust against adaptive attacks, I cannot give the green light to this paper.

[1] Francesco Croce et al., Evaluating the Adversarial Robustness of Adaptive Test-time Defenses, ICML 2022 \
[2] Minjong Lee and Dongwoo Kim, Robust Evaluation of Diffusion-Based Adversarial Purification, ICCV 2023 \
[3] Weili Nie et al., Diffusion Models for Adversarial Purification, ICML 2022

**Questions:**

See the weaknesses above.

---

### Official Review · Reviewer_jmSw · 2023-10-24

**Soundness:** 3 good
**Presentation:** 3 good
**Contribution:** 3 good
**Rating:** 6
**Confidence:** 4

**Summary:**

In this paper, the authors connect adversarial attacks with the diffusion process. They first present how they regard adding adversarial perturbation on clean data as a diffusion process and then propose a reversed adversarial process (RAP) to train a diffusion model to purify the adversarial perturbation as a defense, by which they achieve both better clean accuracy and robustness accuracy compared to the state-of-the-art approaches on Robustbench.

**Strengths:**

	This paper connects adversarial process and diffusion for the first time, and proposes to adversarially train a diffusion adapter to help purify the adversarial perturbations, which is interesting.

	By directly modeling and predicting the adversarial perturbations using the diffusion model, their method can effectively restore the adversarial images to their clean version thus providing a robust defense against adversarial attacks.

	Extensive experiments are conducted to show the proposed method is robust under different adversarial attacks and achieves higher clean accuracy and robustness accuracy compared to the existing state-of-the-art approaches. The results seem solid.

**Weaknesses:**

	The paper does not demonstrate whether the trained diffusion adapter can be generalized to other classifiers.

	Insufficient details or code for reproducibility.

	Although the method works well empirically, the theoretical assumptions lack persuasiveness.

**Questions:**

	Based on my understanding, is it correct that the \tilde(f)_{\theta} in Eq. (7) is just the output of the diffusion model and you do not need to calculate the Jacobian and s^{-t}(x) in practice?
	You may include the detailed derivation for the formula in Sec. 3.1 and 3.2 in the supplementary materials.

	Is the diffusion model pre-trained or trained from scratch?

	One question about the details of generating the adversarial perturbations, which label is used to compute the gradients? The original label of the data or the targeted label you specifically or randomly specified?

	In Sec. 4.5, it is said that the diffusion time step used in your method is 20, is it correct that you also use 20 time steps to denoise the adversarial images? If so, it seems unfair that you only measure 1-step RAP when comparing the total inference time with and without RAP. Could you explain what’s the point of the comparison here?

	Is the proposed method also applicable to adversarial attacks on time series and 3D data?

	You mentioned that “It could be feasible to develop a Foundation Adapter for numerous classifiers”. But just curious, did you conduct experiments to see if the trained adapter in the paper has the capacity to generalize to other classifiers? For example, you train the RAP using wide-resnet-28-10 and replace the classifier with, say, MobileV2. What do you think the results will suggest?

---

### Official Review · Reviewer_ytzM · 2023-10-31

**Soundness:** 2 fair
**Presentation:** 3 good
**Contribution:** 2 fair
**Rating:** 3
**Confidence:** 4

**Summary:**

This paper proposes a novel empirical adversarial defense method against evasion attacks by leveraging the diffusion process as the pre-processing component. Different from the previous works that also leverage the diffusion model as purification, this paper innovatively considers the forward diffusion process as adding perturbation to the inputs. By training a reverse model to continuously remove the perturbation, the model owner can deploy this reverse diffusion processing as a pre-processing defense method to purify the adversarial examples. Empirical results in CIFAR10 against various attacks show the effectiveness of the proposed method.

**Strengths:**

1. The idea of considering the adversarial attack as a diffusion process is novel, and is different from previous works. Previous works leverage the diffusion model as the purifier. Typically, they input the adversarial example into the forward process in the diffusion model and then go through the reverse process to remove the malicious perturbation. Differently, this paper proposes to consider the perturbation injection as the forward diffusion process, and theoretically derive the reverse process. The idea is novel and seems promising.

2. The paper is well-written and easy to understand. This paper successfully distinguished itself from the highly related works. The motivation and the insights are well illustrated by figures.

**Weaknesses:**

The concern is mainly on the evaluation.

1. Since the training and the inference of the diffusion process highly rely on the target attacks, i.e.,  the AutoAttack, it is possible that the defense only works on the attacks that it observed in the training process and the purification may only work for the known attacks. To address this concern, this paper should show experiment results on the transferability/generalizability of the defense. It is better to train on one attack but test on the other attack. In Table 4, the proposed method is trained on $\ell_\infty$ and $\ell_2$ AutoAttacks and tested on $\ell_\infty$ and $\ell_2$ AutoAttacks. The transferability from $\ell_\infty$ and $\ell_2$ or vice versa cannot provide strong evidence for the transferability/generalizability of the proposed method. I would like to see the authors train the diffusion model on some attacks but test it on other totally different attacks.

2. Another concern on the evaluation is about the threat model. Actually, the threat model does not clearly state what the model owner knows about the attacks. One speculation is that the model owner knows about the adversarial example's time step t, or by default, the adversarial example is considered as the middle variable at the T step. This can raise a mismatch problem in the purification. For example, if we generate adversarial attacks with smaller/larger t (smaller/larger perturbation budget), then the adversarial examples may be over-purified or under-purified. And this t is supposed to be unknown by the model owner. I would like to see the authors test the purification on adversarial attacks generated to match different t, after training on adversarial examples generated at T.

3. As a pre-processing defense that can process the image before feeding it to the classifier, the proposed method can be considered as an obfuscation of the gradient of the white-box attacks and thus should be tested using BPDA attacks. It is controversial to generate the adversarial examples using solely the classifier, but test on the pre-processing + classifier bundle. Therefore,  the adversary can try to compute/estimate the gradient of the input through the whole bundle of pre-processing + classifier. If the gradient can not be directly computed, the common practice is to train a deep pre-processing network to simulate the pre-processing process, and then compute the gradient using this deep pre-processing network and the classifier. In this paper, it is not clear how the BPDA+EOT is implemented, thus the results are not convincing, which affects the significance of the proposed method.

**Questions:**

1. Can you provide a full evaluation on the transferability/generalizability?

2. Can you evaluate the diffusion model on the adversarial examples with different perturbation sizes (different t, unknown)?

3. What is the implementation of the BPDA+EOT method?

Ratings may be raised if the concerns are well-addressed.

---

### Official Review · Reviewer_Q6UJ · 2023-11-06

**Soundness:** 3 good
**Presentation:** 3 good
**Contribution:** 3 good
**Rating:** 8
**Confidence:** 4

**Summary:**

The paper aims to give a new defense for adversarial (test time) attacks on ML models using the framework of diffusion decoding. Diffusion and decoding such diffusions are powerful methods for generation of instances, but this paper focuses on using their ideas for defense against adversarial examples.

The idea is to treat the adversarial noise as the diffusion itself and then try to decode it, just like how diffusion models do it for going back to (in that case fresh sampled) instances. What happens here though is that the noise goes away and the instance is classified correctly (in most cases).

The paper tries to extend some of the theories of why diffusion models work, which is based on Gaussian noise, to the general setting. This is done based on assumptions (of invertibility of such noise, which is not true, but is made to get an insight) and is tightly relate to the recent (cited) work of Bansal et al 22) in which the same insight is shared: that diffusion and decoding processes can be done with other types of noise (other than Gaussian) as well; but that work did not focus on using this idea for defenses.

The paper (in my view correctly) draws some analogies between their method and adversarial training. The reason is that here also the attacks that are employed during the training (which in case aim to find the filter that removes the adversarial noise) are done over the partially defended model.

The paper then does some experiments and compare their method with other state of the art methods and show that their method comes out better. One important experiment is also to show that training their filter using attack A does transfer to defending against attack B, which is the core of being adaptive in the analysis if such defenses.

**Strengths:**

The idea of using diffusion for adversarial defense is not new, but the previous work aimed to just use the same Gaussian-denoising, which is at the heart of (default type of) diffusion models to denoise the adversarial noise. But this paper directly aims to remove the adversarial noise itself.  This is a great insight and evidently has worked well (in light of their experiments).

As also mentioned above, I like how the paper does not stop at using an attack type, say A, to both training their filter and test the attack. They do go on to test their filter against

**Weaknesses:**

The theory side of the paper makes an "invertibility" assumption about the adversarial noise, which (as they admit as well) is not realistic. Removing this could be a step forward for that part of the paper.

typos:
"A identified" in page 3 -> As identified

**Questions:**

The paper says in a few places that adversarial noise has a distribution (somehow similar to things like Gaussian, which are distributions). This is not quite accurate. Adversarial "noise" is adaptive, and can depend on the defense. So, it is not a fixed distribution at all. Am I missing something here?

You say that "The outstanding paper (Yoon et al., 2021) provides supporting evidence for the originality of our work." but after reading the quote I still did not get why this is so.

It is true that many adversarial attacks go on in phases and steps, but at the end of the day, adversarial noise is just one piece of noise that one can consider it to be generated in *one* step. Does your method extend to the setting that one is given the *whole* adversarial noise in one piece (as a black-box, in some sense).

Can you explain why Equation 2 is correct? I did not find any explanations.

You say "It is well known that the introduction of randomness can lead to an overestimation of the network’s robustness (Athalye et al., 2018)."
But is not it that Randomized Smoothing is a form of randomization of the test instance, and that it has worked quite well in at least some settings?